# National Early Warning Score (NEWS) Outperforms Quick Sepsis-Related Organ Failure (qSOFA) Score for Early Detection of Sepsis in the Emergency Department

**DOI:** 10.3390/antibiotics11111518

**Published:** 2022-10-31

**Authors:** Dimitri Durr, Tapio Niemi, Jeremie Despraz, Selin Tusgul, Fabrice Dami, Rachid Akrour, Pierre-Nicolas Carron, Marie-Annick Le Pogam, Thierry Calandra, Sylvain Meylan

**Affiliations:** 1Infectious Diseases Service, Service of Immunology and allergy, Department of Medicine, Lausanne University Hospital and University of Lausanne, 1010 Lausanne, Switzerland; 2Department of Epidemiology and Health Systems, Center for Primary Care and Public Health (Unisanté), University of Lausanne, 1015 Lausanne, Switzerland; 3Health Informatics and Data Protection Group, Biomedical Data Science Center, Lausanne University Hospital, 1010 Lausanne, Switzerland; 4Ensemble Hospitalier de la Côte, 1110 Morges, Switzerland; 5Emergency Department, Lausanne University Hospital and University of Lausanne, 1011 Lausanne, Switzerland; 6Service of Geriatric Medicine and Geriatric Rehabilitation, Lausanne University Hospital, 1011 Lausanne, Switzerland

**Keywords:** sepsis, NEWS, quick SOFA (qSOFA) score, prehospital care, emergency department

## Abstract

Background: Prompt recognition of sepsis is critical to improving patients’ outcomes. We compared the performance of NEWS and qSOFA scores as sepsis detection tools in patients admitted to the emergency department (ED) with suspicion of sepsis. Methodology: A single-center 12-month retrospective study comparing NEWS using the recommended cut-off of ≥5 and qSOFA as sepsis screening tools in a cohort of patients transported by emergency medical services (EMS) to the Lausanne University Hospital (LUH). We used the Sepsis-3 consensus definition. The primary study endpoint was the detection of sepsis. Secondary endpoints were ICU admission and 28-day all-cause mortality. Results: Among 886 patients admitted to ED by EMS for suspected infection, 556 (63%) had a complete set of vital parameters panel enabling the calculation of NEWS and qSOFA scores, of whom 300 (54%) had sepsis. For the detection of sepsis, the sensitivity of NEWS > 5 was 86% and that of qSOFA ≥ 2 was 34%. Likewise, the sensitivities of NEWS ≥ 5 for predicting ICU admission and 28-day mortality were higher than those of qSOFA ≥ 2 (82% *versus* 33% and 88% *versus* 37%). Conversely, the specificity of qSOFA ≥ 2 for sepsis detection was higher than that of NEWS ≥ 5 (90% *versus* 55%). The negative predictive value of NEWS > 5 was higher than that of qSOFA ≥ 2 (77% *versus* 54%), while the positive predictive value of qSOFA ≥ 2 was higher than that of NEWS ≥ 5 (80% *versus* 69%). Finally, the accuracy of NEWS ≥ 5 was higher than that of qSOFA ≥ 2 (72% *versus* 60%). Conclusions: The sensitivity of NEWS ≥ 5 was superior to that of qSOFA ≥ 2 to identify patients with sepsis in the ED and predict ICU admission and 28-day mortality. In contrast, qSOFA ≥ 2 had higher specificity and positive predictive values than NEWS ≥ 5 for these three endpoints.

## 1. Introduction

Sepsis is defined as life-threatening organ dysfunction caused by a dysregulated host response to infection [1]. Early recognition and prompt management of sepsis are critical to reducing morbidity and mortality [2]. However, the heterogeneity of clinical manifestations of sepsis hampers the prompt identification of septic patients. Moreover, the latest sepsis definition (Sepsis-3) relies, in part, on laboratory tests, which may potentially delay diagnosis [3]. To address this issue, Seymour and co-workers proposed an abbreviated version of the sepsis-related (or sequential) organ assessment (SOFA) score based on three clinical vital signs coined as quick SOFA (qSOFA) [4]. While its initial validation suggested that qSOFA had similar prediction characteristics as the SOFA score, subsequent studies have shown that qSOFA may have low sensitivity for the detection of patients with severe sepsis as per the Sepsis-2 consensus definitions, which may potentially limit its role as a screening tool [5,6,7,8] The National Early Warning Score (NEWS) [9] is a composite score of seven vital parameters (respiratory rate (RR), heart rate (HR), systolic blood pressure (SBP), pulse oximetry (SpO_2_), temperature (T°), and the Alert-Verbal-Pain-Unresponsive (AVPU) scale) utilized to quantify a patient’s condition. It is one of the most commonly used early warning scores (EWS) worldwide. Some studies suggested that the use of an EWS might be a superior screening tool for sepsis [5,7,8,9] We therefore sought to compare the respective performances of NEWS and qSOFA for the detection of sepsis according to the latest sepsis consensus definitions (Sepsis-3) in a cohort of patients admitted to the ED of the Lausanne University Hospital [6].

## 2. Results

Among 11,411 patients admitted to the ED, 886 had a suspected infection of whom 330 were excluded because of incomplete sets of vital parameters. Among the 556 patients with a complete dataset, 256 (46.0%) had an infection without sepsis and 300 (54.0%) had sepsis according to the Sepsis 3 definitions (Figure 1) [1]. We first looked at the spread of NEWS and qSOFA in the entire study cohort and in the subgroups of patients with infection and sepsis (Figure 2). The respective medians (range) of NEWS and qSOFA scores were 6 (0–17) and 1 (0–3) in the entire cohort, 4 (0–13) and 1 (0–2) in the 243 patients with infection, and 8 (1–17) and 1 (0–3) in the 305 patients with sepsis (*p* value < 0.0001 for both).

For the detection of sepsis in the ED, the sensitivity of NEWS > 5 was 86% (95% CI 82–90%) and that of qSOFA ≥ 2 was 34% (95% CI 29–40%). These numbers were similar if using sepsis-2 definitions, limited to severe sepsis and septic shock (see Appendix A). Likewise, the sensitivities of NEWS ≥ 5 for ICU admission and 28-day mortality were higher than those of qSOFA ≥ 2 [82% (95% CI 70–91%) versus 33% (95% CI 21–47%) and 88% (95% CI 81–93%) versus 37% (95% CI 28–46%)]. Conversely, the specificity of qSOFA ≥ 2 for sepsis detection was higher than that of NEWS ≥ 5 [90% (95% CI 86–94%) *versus* 55% (95% CI 49–62%)]. The negative predictive value of NEWS ≥ 5 was higher than that of qSOFA ≥ 2 [77% (95% CI 70–83%) *versus* 54% (95% CI 49–59%)], while the positive predictive values of qSOFA ≥ 2 were higher than that of NEWS ≥ 5 [80% (95% CI 72–87%) *versus* 69% (95% CI 64–74%)] for sepsis detection, though these remained similar for ICU [13% (95% CI 9–16%) *versus* 15% (95% CI 9–22%)] and 28-day mortality [31% (95% CI 27–36%) *versus* 39% (95% CI 30–48%)]. Finally, the accuracy of NEWS ≥ 5 was higher than that of qSOFA ≥ 2 [72% (95% CI 68–76%) *versus* 60% (95% CI 56–64%)]. ROC curve analyses confirmed the superior performance of NEWS over qSOFA for sepsis detection [0.79 (95% CI 0.74–0.82) *versus* 0.65 (95% CI 0.61–0.69)], ICU admission [0.67 (95% CI 0.59–0.74) *versus* 0.58 (95% CI 0.51–0.65)], and mortality [0.70 (95% CI 0.65–0.75) *versus* 0.62 (95% CI 0.57–0.67)] (Figure 3).

We then analyzed the relative contribution of each vital sign to AUCs for the detection of sepsis by decomposing the relative contributions of vital parameters for NEWS to receiver operating characteristics (ROC) curve analyses. For the entire patient cohort, FiO_2_ and SpO_2_ had the most impact (0.11 and 0.10 relative contributions) on the NEWS performance for the detection of sepsis (Figure 4). Given their potential impact on various vital parameters, we assessed whether age or site of infection influenced the performance of NEWS. Neither influenced the performance of NEWS in a significant manner (Appendix A). 

## 3. Discussion

We found that a NEWS score equal to or greater than 5 had higher sensitivity than a qSOFA score equal to or greater than 2 to identify patients with sepsis admitted to the emergency department of a large university medical center in the Western part of Switzerland. NEWS was also found to be superior to qSOFA for the prediction of ICU admission and 28-day mortality. On the contrary, qSOFA was more specific than NEWS and it exhibited a higher positive predictive value than NEWS. The latter showed a higher negative predictive value. Taken altogether, the present findings suggest that NEWS would be the favored score for the detection of sepsis in the emergency department, while qSOFA would be a more specific sepsis detection tool with the capacity to reduce the rate of false positives.

To the best of our knowledge, this is the first study comparing NEWS and qSOFA using the latest international consensus definitions for sepsis and septic shock (Sepsis-3) [1] Previous studies have compared NEWS and qSOFA using severe sepsis according to the second consensus definitions (now labelled Sepsis-2) or endpoints such as mortality [5,7] When using Sepsis-2 definitions, results were similar, although, in both instances, diagnosis relied on discharge summaries for severe sepsis [5,7]. Such studies may be fraught by the quality of sepsis diagnosis as suggested by a report showing underreporting with sepsis documented in less than half of the death certificates of septic patients [11] Given that we used the criterion of a change in SOFA score to identify patients with sepsis, the present study is less prone to such a reporting bias.

In the present study, we used a NEWS cut-off value of 5 or higher, as is applied in England for the early detection of sepsis in the emergency department (ED) [9] Mathematically, a qSOFA of 2 may correspond to a NEWS of 4 (e.g., systolic blood pressure of 100 mmHg and respiratory rate of 22 breaths per minute with other parameters at 0 points). A NEWS ≥ 4 would increase the sensitivity to 91% of cases Nevertheless, a NEWS ≥ 5 identifies 86% of cases, missing 42 of a total of 300 patients with S3 sepsis in our case. However, several factors must be weighed in establishing a cut-off. First, while it may increase sensitivity, a NEWS threshold of 4 may cause alert fatigue. This is particularly true in patients with chronic conditions such as chronic obstructive pulmonary diseases requiring home oxygen. Conversely, a NEWS threshold at 5 may be less prone to alert fatigue, but it may miss a fraction of patients with a qSOFA of 2—in our case, only one patient—(hypotension and increased respiratory rate) with an increased risk of mortality.

In contrast to NEWS, a score designed to identify patient deterioration irrespective of its etiology, qSOFA, was specifically designed to identify infected patients at increased risk of mortality [4] Consistent with the feature of qSOFA, the specificity and the positive predictive value of qSOFA were superior to NEWS. Our results support the recommendation of the 2021 guidelines of the Surviving Sepsis Campaign (SSC) to use NEWS and not qSOFA as a single screening tool for the detection of sepsis and septic shock [2]. The present findings also suggested the possibility to include NEWS as an alarm system in electronic health records [12]. Given its excellent positive predictive value, qSOFA may accelerate the management of septic patients. As an example, the 2021 international guidelines of the Surviving Sepsis Campaign for the management of sepsis and septic shock distinguish “high-likelihood” and “possible” sepsis for the timing of antibiotics administration (1 h *versus* 3 h, respectively) [2]. In view of its high positive predictive value, the qSOFA ≥ 2 may help clinicians to identify the most severely ill patients in whom prompt initiation of antibiotics is required. The present findings would support the use of a two-level strategy with NEWS ≥ 5 for the prompt identification of sepsis patients and qSOFA for the identification of the sickest sepsis patients.

Our work has several limitations. First, 38% of cases were excluded due to missing parameters for the calculation of NEWS or qSOFA. Second, missing bilirubin values (91% of cases) were deemed normal, which may result in an underestimation of the number of sepsis cases. The third limitation is that NEWS is a score of patient deterioration; here, we compare sensitivity amongst a cohort of infected patients. However, this may not capture the performance of NEWS in an emergency department with non-infected patients in addition to infected patients.

## 4. Material and Methods

### 4.1. Study Design, Setting, and Participants

We used a previously described cohort of patients brought by EMS to the LUH emergency department [6]. Among 11,411 patients admitted to the ED, 886 patients (7.8%) fulfilled the criteria of infection defined as an infection contracted outside the hospital and diagnosed within 48 h of hospital admission. Sepsis was defined as a life-threatening organ dysfunction caused by a dysregulated host response to an infection [1] (Figure 1).

### 4.2. Score Computation

We computed SOFA scores as described previously with the following assumptions or corrections [1,13] First, in the absence of bilirubin (87% of cases), we assumed that the value was less than 20 mmol/L [14]. When PaO2 was not available, pulse oximetry values (SpO2) were used (Appendix A). As sepsis diagnosis is based on infection with a rise in a patient’s SOFA score, we sought to identify patients with a chronic disease causing a higher SOFA score unrelated to sepsis. Codes from the 10th revision of the International Classification of Disease (ICD-10—a French translation of ICD-10-GM used in Switzerland) were browsed for chronic diseases. All codes containing the word ‘chronic’ but lacking the word ‘acute’ were selected. The list of codes was then refined for codes specific to organs associated with the SOFA score (Circulation: I, Coagulation: D65-–D69, Renal: N0–N3, Liver: K7, Nervous: G, Respiration: J). When an ICD-10 code was present in the discharge summary of the patient stay, the SOFA component with a matching chronic disease was ignored when computing the final score (e.g., creatinine value not considered for SOFA computation in chronic renal failure patients). Patients were allocated to the sepsis group according to a SOFA score equal to or greater than 2 points. The NEWS score was computed as described previously [15] The need for supplemental oxygen was assessed according to the FiO2 given by the arterial blood gases (ABG) analysis in the ED. In the same way, we converted the Glasgow Coma Scale (GCS) to the AVPU scale according to the validated scale conversion method [12]. We elected to use a threshold of NEWS equal to or greater than 5 (≥ 5) and a qSOFA equal to or greater than 2 for the comparison of the performance of NEWS and qSOFA for the detection of sepsis and for the prediction of ICU admission and 28-day all-cause mortality (Table 1).

### 4.3. Statistical Analyses

The statistical analyses were performed using a random Forest/Z-value method. Statistical metrics and their 95% confidence intervals were computed using the exact method of the epi.tests function in R’s epiR package (epiR: Tools for the Analysis of Epidemiological Data. R package version 2.0.19. https://CRAN.R-project.org/package=epiR, accessed on 29 September 2022). We used conditional permutation importance [10] for random Forest models to estimate the variable importance. The importance value is the mean decrease in the AUC of the model if the variable was not included in the model. If the value is zero (or even negative), the variable does not improve the AUC of the model at all. For example, if the AUC of the model is 0.80 and the importance of a variable is 0.09, a model without this variable would have an AUC value 0.71. The random forest model was fitted using the random Forest function of the random Forest package [16]. AUC values for the random forest model were computed using R’s pROC package [17].

## 5. Conclusions

NEWS has better sensitivity, accuracy, and negative predictive value than qSOFA for the early detection of sepsis in an emergency department setting. Conversely, qSOFA has excellent positive predictive value. The use of both scores for the detection of sepsis (NEWS) and the rapid assessment of the severity of sepsis at the bedside (qSOFA) may best serve the evaluation and management of patients with sepsis at the time of admission in the emergency department.

## Figures and Tables

**Figure 1 antibiotics-11-01518-f001:**
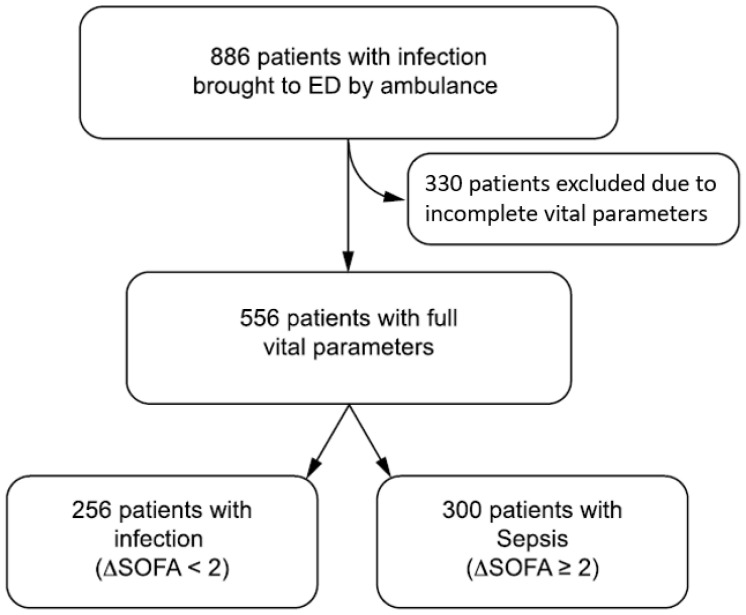
Study flowchart.

**Figure 2 antibiotics-11-01518-f002:**
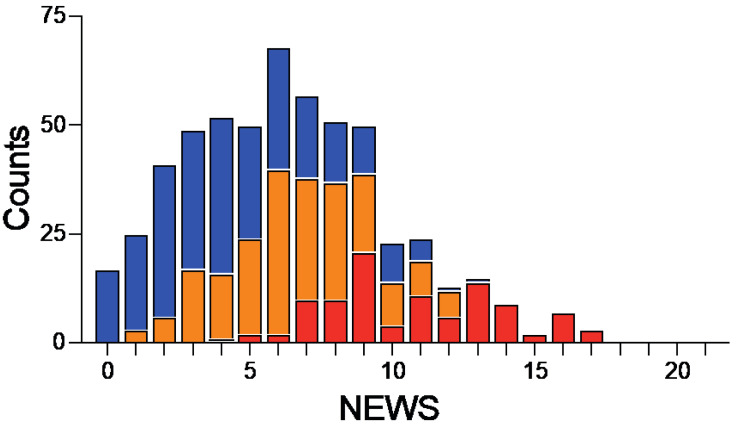
**Distribution of patients with infection and sepsis according to NEWS.** Number of patients with infection without sepsis (blue bars) and sepsis with a qSOFA < 2 (orange bars) or a qSOFA ≥ 2 (red bars) for each category of NEWS.

**Figure 3 antibiotics-11-01518-f003:**
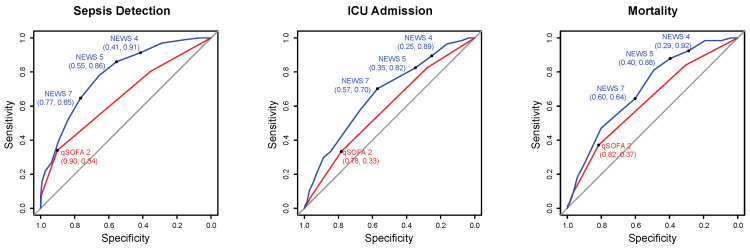
Receiver operating characteristic (ROC) curves for sepsis detection, ICU admission, and 28-day mortality for NEWS and qSOFA scores. ROC curves of NEWS (blue) and qSOFA (red). The area under the ROC curves (AUC) for NEWS and qSOFA were 0.789 (95% CI 0.75–0.82) vs. 0.65 (95% CI 0.61–0.69) for sepsis detection; 0.67 (95% CI 0.60–0.744) vs. 0.58 (95% CI 0.51–0.68) for ICU admission and 0.70 (95% CI 0.65–0.75) vs. 0.63 (95% CI 0.58–0.68) for 28-day mortality.

**Figure 4 antibiotics-11-01518-f004:**
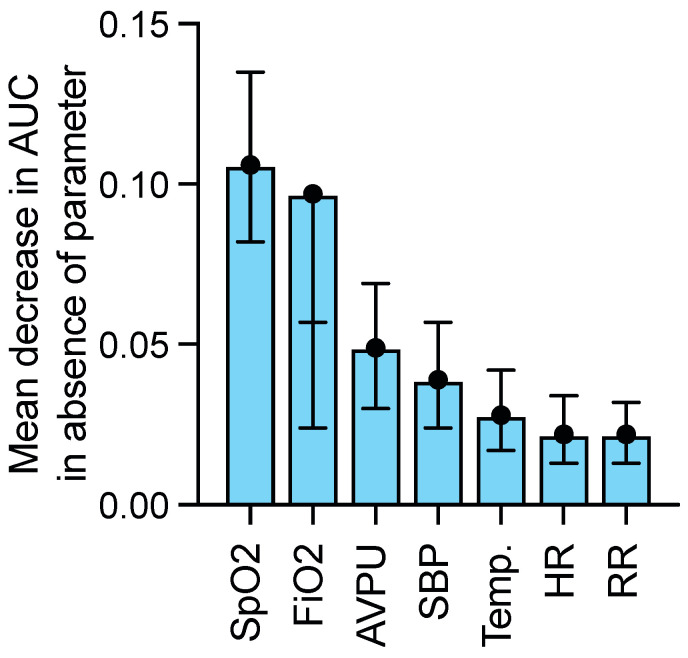
**Relative contribution of each variable of NEWS for the detection of sepsis.** Conditional permutation importance [10] for random forest models was used to estimate the relative contribution of each variable of NEWS to the area under the curve of the model. Relative contribution of each variable: O2 saturation (SpO2) 0.106, fractionated inspired O2 (FiO2) 0.097, alert/verbal/pain/unresponsive mental state (AVPU) 0.049, systolic blood pressure (SBP) 0.057, temperature (Temp.) 0.028, heart rate (HR) 0.034, respiratory rate (RR) 0.022.

**Table 1 antibiotics-11-01518-t001:** Sensitivity, specificity, positive and negative predictive values of NEWS and qSOFA for sepsis detection, ICU admission, and 28-day mortality.

		NEWS ≥ 5				qSOFA ≥ 2			
	N	Sensitivity (95% CI)	Specificity (95% CI)	Positive Predictive Value	Negative Predictive Value	Sensitivity (95% CI)	Specificity (95% CI)	Positive Predictive Value	Negative Predictive Value
**Sepsis**	**300** **(54.0%)**	86 (82–90)	55 (49–62)	69 (64–74)	77 (70–83)	34 (29–40)	90 (86–94)	80 (72–87)	54 (49–59)
**ICU admission**	**57** **(10.3%)**	82 (70–91)	35 (31–39)	13 (9–16)	95 (90–97)	33 (21–47)	78 (74–82)	15 (9–22)	91 (88–94)
**28-day mortality**	**132** **(23.7%)**	88 (81–93)	40 (35–44)	31 (27–36)	91 (86–95)	37 (29–46)	82 (78–85)	39 (30–48)	81 (77–84)

## Data Availability

The datasets used and/or analyzed during the current study are available on the Zenodo platform (https://zenodo.org/deposit/7244112 accessed on 29 September 2022).

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
