# Peer review of "National Early Warning Score (NEWS) Outperforms Quick Sepsis-Related Organ Failure (qSOFA) Score for Early Detection of Sepsis in the Emergency Department"

_antibiotics, 2022, doi:10.3390/antibiotics11111518_

Round 1
Reviewer 1 Report
Thank you for the opportunity to review this manuscript. This retrospective study included ED patients within one year. The authors concluded that the NEWS performed better than qSOFA in sepsis identification as well as risk of ICU admission and mortality. I have several comments that authors should address before it can be published:
1. The more direct method to compare two screening tools in discriminative ability maybe ROC curve? Please provide the ROC curve as well as AUC of NEWS/qSOFA in sepsis detection, ICU admission and mortality risk. Meanwhile, the optimal cut-off value should also be provided.
2. It would be interesting to compare NEWS score with the widely-used risk stratification score, such as SOFA and APACHEII score.
3. This is a retrospective study. A validation cohort in a prospective method is suggested to confirm these findings.
4. Line 173. 2021 SSC guidelines only recommend against using qSOFA as a single screening tool for sepsis/septic shock. They do not mentioned of any recommendation regarding the using of NEWS since different screening tools have wide variety of sepsis detection sensitivity and specificity. Please re-phrase these sentences to avoid mis-understanding.
Reviewer 2 Report
diagnosis of sepsis in the emergency department.
The manuscript deserves several modifications before a possible acceptance for publication.
I think that the presence of a microbiologist could be of major interest to add in the authors, sepsis being (also) based on (micro)biological criteria
The highlights are not necessary or are not sufficiently highlighted.
Bibliographic references are not standardized as recommended.
Prefer passive forms.
Italicize "e.g."/"versus".
Subtitles are not formatted as recommended.
epiR package must be referenced appropriately
I think it might be interesting to take advantage of the dataset to compare the scores according to the sepsis-3 and also sepsis-2 definition to strengthen the robustness of the study.
In the presentation of the results, the threshold was determined arbitrarily, which is not acceptable and should be subject to ROC/AUROC curve analysis.
A Venn diagram would be interesting to understand the proportions of discordant results. The authors need to discuss this better.
The authors should discuss the fact that the PPVs are not different between the scores tested.
It is not acceptable that the definition of diastolic pressure is assessed subjectively (Line 186). Please consider re-analysis.
Due to the lack of data on an important parameter of sepsis, bilirubin, the authors should consider re-analyzing the studies by excluding these patients, or by performing a robustness analysis.
Data used and/or analyzed during the study should be made available and not "available upon reasonable request to the corresponding author".
Round 2
Reviewer 1 Report
thanks for the author's reply.
I have no further comments.
Reviewer 2 Report
The manuscript has been appropriately revised according to my previous comments and is now, to my opinion, suitable for publication.